# A drop dispenser for simplifying on-farm detection of foodborne pathogens

**Mohsen Ranjbaran[1,2], Simerdeep Kaur[1,2], Jiangshan Wang[1,2], Bibek Raut[3], Mohit S. Verma[1,2,3]***

1 Department of Agricultural and Biological Engineering, Purdue University, West Lafayette, Indiana, United States of America, 2 Birck Nanotechnology Center, Purdue University, West Lafayette, Indiana, United States of America, 3 Weldon School of Biomedical Engineering, Purdue University, West Lafayette, Indiana, United States of America

* msverma@purdue.edu

**Data Availability Statement:** All relevant data are within the paper and its Supporting Information files.

**Funding:** The work presented here is funded by CPS AWARD NUMBER: 2021CPS12, CDF

## Abstract

Nucleic-acid biosensors have emerged as useful tools for on-farm detection of foodborne pathogens on fresh produce. Such tools are specifically designed to be user-friendly so that a producer can operate them with minimal training and in a few simple steps. However, one challenge in the deployment of these biosensors is delivering precise sample volumes to the biosensor's reaction sites. To address this challenge, we developed an innovative drop dispenser using advanced 3D printing technology, combined with a hydrophilic surface chemistry treatment. This dispenser enables the generation of precise sample drops, containing DNA or bacterial samples, in volumes as small as a few micro-liters ($\sim 20$ to $\sim 33$ μL). The drop generator was tested over an extended period to assess its durability and usability over time. The results indicated that the drop dispensers have a shelf life of approximately one month. In addition, the device was rigorously validated for nucleic acid testing, specifically by using loop-mediated isothermal amplification (LAMP) for the detection of *Escherichia coli* O157, a prevalent foodborne pathogen. To simulate real-world conditions, we tested the drop dispensers by integrating them into an on-farm sample collection system, ensuring they deliver samples accurately and consistently for nucleic acid testing in the field. Our results demonstrated similar performance to commercial pipettors in LAMP assays, with a limit of detection of $7.8 \times 10^6$ cells/mL for whole-cell detection. This combination of precision, ease of use, and durability make our drop dispenser a promising tool for enhancing the effectiveness of nucleic acid biosensors in the field.

## 1. Introduction

Current standard lab-based methods of nucleic acid testing for foodborne pathogens in fresh produce involve enrichment and detection by quantitative polymerase chain reaction (qPCR), which is slow and could require more than 18 hours [1]. The sample enrichment step prevents us from obtaining quantitative information about the abundance of bacteria of interest [2]. In addition, the produce samples need to be transported from the farm to the lab which adds

Agreement No: 20-0001-054-SF USDA Cooperative Agreement No. USDA-AMS-TM-SCBGP-G-20-0003. Any opinions, findings, conclusions, or recommendations expressed in this publication or audiovisual are those of the author(s) and do not necessarily reflect the views of The Center for Produce Safety (CPS), the California Department of Food and Agriculture, or the Agricultural Marketing Service of the U.S. Department of Agriculture (USDA). The work upon which this project entitled "Field evaluation of microfluidic paper-based analytical devices for microbial source tracking" was funded in whole or in part through a subrecipient grant awarded to CPS through the California Department of Food and Agriculture 2020 Specialty Crop Block Grant Program and the USDA's Agricultural Marketing Service. The funders had no role in study design, data collection and analysis, decision to publish, or preparation of the manuscript. The funding organization (CPS) reviewed the manuscript prior to submission to ensure the accuracy of the information and address any concerns about the publication of sensitive content.

**Competing interests:** M.S.V. has an interest in Krishi Inc., which is a startup that is interested in commercializing technologies developed here. This work was not funded by Krishi Inc. This conflict does not alter our adherence to PLOS ONE policies on sharing data and materials.

delays in microbial detection, and therefore in decision-making about growing and harvesting. Instead, isothermal DNA amplification methods, such as loop-mediated isothermal amplification (LAMP) could be used for on-farm bacterial detection [3–10]. However, the poor user-friendliness of available LAMP biosensors is a major barrier against their application for on-farm detection [11]. Improving user-friendliness partly requires equipping the biosensors with a reliable sample delivery device [12]. This device would measure a prescribed volume of an aqueous sample obtained from the surface of the produce, collection flags, or harvesters and deliver it to the LAMP reaction sites [6]. The fluid-delivery mechanism needs to be easy to use so that a producer or other non-specialists can operate it without requiring several preparation steps [13].

A number of studies have reported the application of LAMP for the detection of foodborne pathogens [14–21] on fresh produce and/or plant materials. A paper-based device combining DNA extraction and DNA amplification (using LAMP) was used for colorimetric detection of *Escherichia coli* (ATCC:25922) in several media, including spinach leaf extracts, with a limit of detection of $1\times10^3$ CFU/mL [22]. In another study, a fast and specific LAMP assay for the detection of *Xanthomonas fragariae* on strawberry leaves was developed [23]. The assay time was 7–20 min with a detection limit of $1\times10^2$ CFU/mL, highlighting the applicability of LAMP for on-site microbial screening of plant materials. While attention was mainly devoted to the development of molecular assays for the detection of certain microbial species, more effort is needed to reliably run these assays in the field [24].

One of the bottlenecks in the development of fully integrated sample-to-answer nucleic-acid biosensors for on-farm applications is the delivery of a prescribed volume of DNA samples to the reaction sites [25–27]. So far, many microfluidic sample delivery devices have been introduced with the potential for on-site nucleic-acid testing, including those used for clinical [28–35] and food safety [36–42] applications. However, the majority of these devices are not suitable for nucleic acid tests. Low-end devices such as capillary micropipettes often lack the necessary accuracy, while high-end devices like lab pipettes are costly and not easily accessible outside laboratory environments. Keeping the importance of user-friendliness and cost efficiency in mind, we designed a device for which the user will only need to press a button to deliver a measured amount of liquid to the downstream reaction sites. In the future, the device can be improved by integrating sample preparation and multiplexed dispensing into a single device. This device is part of a broader effort to develop a user-friendly biosensor for on-farm detection of fresh produce microbial contamination [4–6, 43–46]. The device was used for sample-to-answer detection of Shiga-toxin-producing *E. coli* (STEC) O157:H7 obtained from the swabbing of artificially contaminated collection flags at various microbial concentrations. This device could be part of a simple consumable kit that would cost approximately USD 4 per test.

## 2. Materials and methods

### 2.1 Design and fabrication of the drop dispenser

All computer-aided designs were performed in the Fusion 360 software (Autodesk, CA). Fig 1A shows a cross-section of the drop dispenser. The device consists of two main parts: a plunger that moves up and down when the user operates it, and a liquid holder with a capillary tube at the discharge. After designing the device in Fusion 360, the Standard Triangle Language (.stl) files were exported and transferred to a Form 3B stereolithography 3D printer (Formlabs, MA) to fabricate the 3D-printed devices. Different resin types were used and tested during 3D printing including High Temp V2 (Formlabs, RS-F2-HTAM-02), Rigid 4000 V1 (Formlabs, RS-F2-RGWH-01), BioMed Clear V1 (Formlabs, RS-F2-BMCL-01), and Clear V4

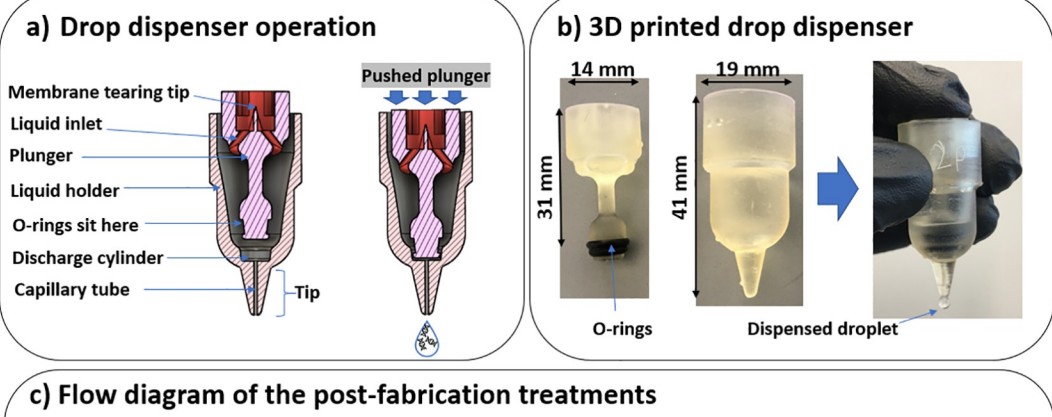

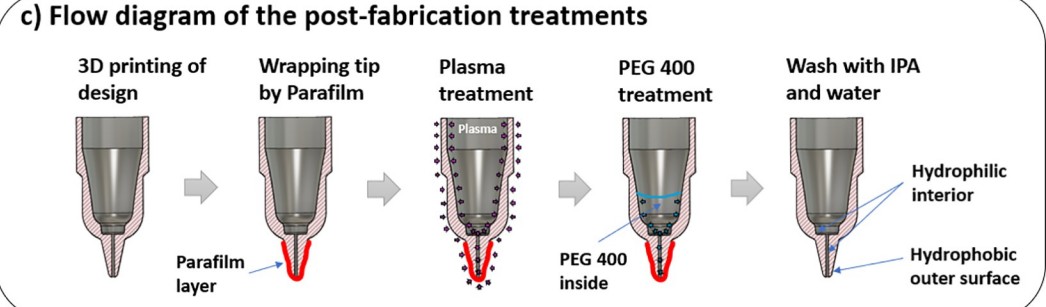

**Fig 1. Design, fabrication and operation of the drop dispenser.** a) Cross section of the drop dispenser with description of the components. b) A typical 3D printed device with drop generation. c) A flow diagram of the post-fabrication surface treatment of the liquid holder by plasma and PEG 400 exposure.

resins (Formlabs, RS-F2-GPCL-04). In order to develop devices of different drop volumes, various design parameters were changed. Devices with different tip lengths, capillary tube diameters, discharge cylinder depths, and tip diameters and angles were fabricated using the selected resin and tested. The fabricated devices were then rinsed using isopropyl alcohol (IPA) for 30 min and were cured by exposing them to UV light at 55˚C for 30 min. Fig 1B shows the two 3D-printed parts as well as two O-rings (Helipal, Airy-Acc-Oring-2.5×6 mm) that are used to provide sealing. When the plunger is pushed down, the O-rings seal the discharge cylinder from the liquid holder. Further downward movement of the plunger causes the liquid within the discharge cylinder and the capillary tube to be dispensed out. At this stage, the device acts as a positive displacement pump. Although we did not use it in the devices for our experiments, a compression spring (0.6 mm×9.5 mm×20 mm) installed at the plunger neck, could be used to facilitate the back movement of the plunger after its release.

## 2.2 Surface treatment and post-wash of the drop dispenser

We investigated a number of surface treatment approaches to enhance the reliability of the drop dispenser. These approaches included exposure to plasma for the different duration (2 or 4 min), and exposure to polyethylene glycol (PEG) 400 (Fisher Scientific, P167-1) for 24 h after plasma treatment by either dipping the device in the PEG 400 or only pouring 600 μL of PEG 400 into the interior of the device. To perform the latter approach (Fig 1C), we wrapped the tip of the liquid holder with multiple layers of Parafilm wrapping film (Fisher Scientific, S37440) before exposing it to plasma and PEG 400. The plasma treatment was performed using an air plasma generator (Plasma Etch, Inc., PE-25) at about 0.2 Torr (26.7 Pa). This treatment approach kept the outer tip surface hydrophobic but made the inner capillary tube hydrophilic.

After surface treatment, the assembled drop dispenser—including the plunger, O-rings, and the liquid holder—was thoroughly washed with IPA for 10 min and rinsed with plenty of ultra-pure water to remove any residue that could interfere with LAMP reactions. The clean devices were dried using an air gun and stored in separate reclosable Polypropylene bags (Uline, S-17954) for future use.

## 2.3 Precision tests of the drop dispenser

The efficacy of the drop dispenser was tested at various stages of development. Devices made from multiple resins (discussed above) were initially tested and compared to select the most appropriate resin that led to the highest precision of drop generation. For this purpose, 4–6 devices from each resin were fabricated and tested for the generation of 30 drops. After selecting the most appropriate resin, 6 devices were fabricated out of it and used for over-time precision tests. Each device was tested 0, 3, 7, 10, 14, 17, 21, 28, and 35 days after PEG 400 treatment. Between the tests, the devices were stored at room temperature in specific boxes. For each test, the device was used to generate 30 drops. The mass of each drop was measured using a Mettler Toledo mass balance (Fisher Scientific, 01-912-402)—with an accuracy of $\pm 0.0001$g—and converted to volume units, assuming a water density of 1000 kg/m$^3$. After each test, the device was dried using an air gun and stored until the next experiment. The precision tests continued until at least one of the devices failed by showing uncontrolled dripping after multiple applications. In addition, we also tested the precision and reproducibility of drop generation of a number of commercial micro-pipettors (shown in S1 Fig in S1 File) to compare with our devices.

## 2.4 Bacterial strains, culturing, and quantification

The bacterial strain, *Escherichia coli* (Migula) Castellani and Chalmers (ATCC® 35150™) was cultured overnight in 3 mL of brain heart infusion (BHI) growth medium (VWR International, LLC, 95021–488) using a shaker-incubator at 37°C. To provide a subculture, a 3 μL of the culture was again transferred to a 3-mL fresh BHI medium and incubated for another 16 hr. Then, the bacteria were either used for DNA extraction, or the dilutions were used to artificially contaminate the collection flags, as described in the methods section of "LAMP assays on samples from artificially contaminated collection flags using drop dispenser".

To enable bacterial counting using optical density measurement at 600 nm (OD$_{600}$), a calibration curve was prepared. For this purpose, 1 mL of the initial bacterial culture, in the BHI broth was centrifuged at 9000 rpm for 1 min. After removing the supernatant liquid, the cells were resuspended and well-mixed in the same amount of water and centrifuged again. After a second resuspension in the same amount of water, several serial dilutions with dilution factors of 1, 2, 10, 50, and 100 were prepared in three replicates using molecular biology-grade water. For each dilution, the OD$_{600}$ was read using a CLARIOstar microplate reader (BMG Labtech, Germany). The same dilutions were immediately used to count the cells using a QUANTOM Tx microbial cell counter (Logos Biosystems, South Korea), following the manufacturer's protocols for total bacterial counting (QUANTOM Total Cell Staining Kit, Q13501). The calibration curve and corresponding data are shown in S2 Fig in S1 File.

## 2.5 DNA extraction, purification, and quantification

The DNA extraction was performed using the Invitrogen PureLink Genomic DNA Mini Kit (Fisher Scientific, K182001). 1 mL of overnight bacterial culture was centrifuged to harvest the cell pellet. The cell pellet was re-suspended in 180 μL PureLink Genomic Digestion Buffer. Then, 20 μL of Proteinase K was added to lyse the cells. After vortexing briefly (for about 1

min), the tube was incubated at 55˚C to complete cell lysis. A homogeneous mix of the lysate was obtained after adding and vortexing 20 μL RNase A, 200 μL PureLink Genomic Lysis/ Binding Buffer, and 200 μL 96–100% ethanol. 640 μL of the lysate was added to a PureLink Spin Column and centrifuged at 10,000×g for 1 minute at room temperature. Then the spin column was placed into a clean PureLink collection tube. 500 μL of Wash Buffer 1 was added to the column and the column was centrifuged at 10,000×g for 1 minute at room temperature. Then, the spin column was placed into a clean PureLink collection tube. 500 μL of Wash Buffer 2 prepared with ethanol was added to the column and the column was centrifuged for 3 minutes at room temperature. The DNA was eluted by placing the spin column in a sterile 1.5-mL microcentrifuge tube. Then 30 μL of PureLink Genomic Elution Buffer was added to the column and the column was incubated for 1 min and centrifuged at maximum speed (14,000 rpm) for 1 min. This elution step was repeated twice and the DNA was gathered in a 1.5-mL microcentrifuge tube and stored at -20˚C until used for experiments.

To quantify DNA concentration, 50 μL of serial dilutions (starting from 2 ng/μL, we performed a series of eight dilutions in a 1:2 ratio) of synthetic DNA (i.e., Lambda DNA; Thermo Fisher Scientific, SD0011) were prepared using Invitrogen 1× TE buffer (Fisher Scientific, 12-090-015). These dilutions are shown in S3A Fig in S1 File. Also, using 5 μL of the genomic DNA, 50 μL serial dilutions (began with a DNA stock and performed a series of dilutions, starting with two 1:10 dilutions, followed by four 1:2 dilutions) were prepared (S3B Fig in S1 File). 50 μL of diluted Invitrogen PicoGreen dye (Fisher Scientific, P11496), which was a mix of 6.5 μL dye in 1293.5 μL water, was added to all samples and incubated for 5 min at room temperature. Using a 96-well PCR plate, 25 μL of each DNA sample was added to 3 wells. After sealing the plate using a PCR film, the plate was incubated in a thermocycler at 25˚C. After incubating for 3 minutes at a constant temperature, the PicoGreen fluorescent intensity data were retrieved. Using linear regression, a trendline was obtained for the intensity versus the concentration range of the Lambda DNA (S3A Fig in S1 File). Then, the equation of the trendline was used to estimate the concentration of the extracted genomic DNA which was 227.9 ng/μL (S3B Fig in S1 File).

## 2.6 LAMP reaction mix preparation

By revising the available protocols for LAMP mix preparation [43, 46], we developed a homemade 4× LAMP mix which consists of 100 μL KCl (1000 mM; Sigma-Aldrich, P9541), 160 μL MgSO$_4$ (100 mM; Sigma-Aldrich, M2773), 112 μL Deoxynucleotide triphosphate (dNTP) (25 mM; Fisher Scientific, FERR0182), 2.8 μL Deoxyuridine Triphosphate (dUTP) (100 mM; Fisher Scientific, FERR0133), 0.4 μL Antarctic Thermolabile UDG (1 U/μL; New England Biolabs, M0372S), 5.4 μL Bst2.0 DNA Polymerase (120 U/μL; New England Biolabs, M0537M), 20 μL phenol red solution (25 mM; Sigma-Aldrich, P3532), and 99.4 μL nuclease-Free water (Fisher Scientific, 43-879-36). After mixing, the pH was adjusted to about 7.8–7.9 using KOH (0.1 M or 1 M) leading to a red but not pink solution. The pH measurements were performed by a micro-pH electrode (Fisher Scientific, 11-747-328). To prepare the 20× LAMP primer mix, the primer sets shown in S1 Table in S1 File [47] acquired from published sequences were mixed by adding 80 μL FIP, 80 μL BIP, 20 μL LF, 20 μL LB, 10 μL F3, 10 μL B3 (each 100 μM), and 30 μL nuclease-free water. The primers were synthesized by Life Technologies and purified using a desalting process. After mixing, the primer mix was heated at 95˚C for 10 min prior to usage. The final LAMP master mix (about 9 μL per reaction) was prepared by using 7.5 μL of the 4× mix, 1.5 μL of 20x primer mix, and 0.12 μL of betaine (5 M; Sigma-Aldrich, B0300-5VL). Then, about 21 μL of template—either extracted DNA dilutions (methods section "LAMP assays on extracted bacterial DNA using drop dispenser") or resuspended bacterial

culture (methods section "LAMP assays on samples from artificially contaminated collection flags using drop dispenser")—was added using either an Eppendorf 20–200 μL pipettor or our drop dispenser.

## 2.7 LAMP assays on extracted bacterial DNA using drop dispenser

In order to investigate the efficacy of the drop dispenser in running LAMP assays, as compared with commercial pipettors, we conducted experiments to measure the limit of detection (LoD) in each condition. Here, we used six levels of *E. coli* O157:H7 DNA concentration, including 10,000, 5000, 2500, 1250, 500, and 250 copies per reaction, as well as negative controls. For each level, three replicates were considered, and the LoD tests were conducted three times on different days to ensure the results were repeatable. Each assay was performed in a 0.2-mL PCR tube. Therefore, we ran a total of 42 reactions per LoD test. After preparation of all reaction mixes and adding the DNA templates (following the protocol described in the method section "LAMP reaction mix preparation"), they were placed in 3D-printed tube holders— made from a Rigid 4000 V1 resin (Formlabs, RS-F2-RGWH-01) with white color—and submerged in a water bath for 60 min at 65˚C. The temperature of the water bath had been verified previously using a Hti HT-04 Thermal Imaging Camera [4]. The colorimetric LAMP assay was considered positive if the color of the reaction mix turned yellow at the end of the heating process. Otherwise, a red color indicated a negative result.

## 2.8 LAMP assays on samples from artificially contaminated collection flags using drop dispenser

The drop dispenser developed in this study will be a part of a larger kit that is intended to be used for running on-farm LAMP assays using crude samples. Therefore, we evaluated its efficacy using bacterial cells—instead of just extracted DNA—as the template. In our recent work [6], we demonstrated that the use of collection flags (made from plastic sheets) is better for the collection of bioaerosols as compared to lettuce leaves. These collection flags provide a more consistent and reproducible result most likely due to the ease of swabbing flat plastic compared to leaves with grooves. Thus, we fabricated and used collection flags for the LAMP assays. Each collection flag consists of one piece (5 cm × 30 cm) of a transparency film (Apollo Plain Paper Copier Transparency Film, 617993) attached to a wooden stick. For the purpose of the current in-lab experiments, we did not attach the flags on the sticks.

Five dilution levels of the bacterial culture (method section "Bacterial strains, culturing, and quantification")—with dilution factors of 1, 10, 100, 1000, and 10000—were prepared. For this purpose, 2 mL of the initial bacterial culture in the BHI broth was centrifuged at 9,000 rpm for 1 min. After removing the supernatant liquid, the cells were resuspended and well-mixed in the same amount of water and centrifuged again. After a second resuspension in the same amount of water, the $OD_{600}$ was read using a CLARIOstar microplate reader (BMG Labtech, Germany) and the resuspended bacteria were used to generate further dilutions. The $OD_{600}$ was translated into bacterial count using a calibration curve (S2 Fig in S1 File). In addition to the bacterial dilutions, water and bacterial DNA extract ($1 \times 10^5$ copies/μL) were also used as negative and positive control templates, respectively.

200 μL of each template (bacterial dilutions, bacterial DNA, and water) was spot inoculated on one side of a collection flag. For each template, we used three clean flags laid inside a biosafety cabinet. Using separate sterile inoculating loops, the inoculums were gently spread over the entire surface of each flag and left to dry out for 60 min. Then, using a wet polyester-tipped swab (BD BBL, 263000), the templates were collected from each flag surface. To do that, a sterile swab was first dipped in molecular biology-grade water. Then, the wet swab was rubbed

over the entire surface of the flag and finally resuspended in 200µL molecular biology-grade water. The samples were transferred to drop dispensers, which were used to add a precise volume of sample for running LAMP assays in 0.2-mL PCR tubes.

## 3. Results

### 3.1 Fabrication material selection and dispensing capacity

Several drop dispensers were initially fabricated out of High Temp V2 resin and tested to determine the best surface treatment approach. Devices without any surface treatment could not hold the liquid within the liquid holder. Generally, plasma treatment was effective to improve the holding of the liquid within the device. However, drop generation was only possible for those devices whose tips were already wrapped with a Parafilm wrapping film layer during plasma treatment (Fig 2A). Devices treated with a 2-min plasma exposure better-dispensed drops than those with a 4-min exposure and provided less variability (Fig 2B). We tested the 2-min plasma-treated devices twice with 10 days intervals and observed that the precision of

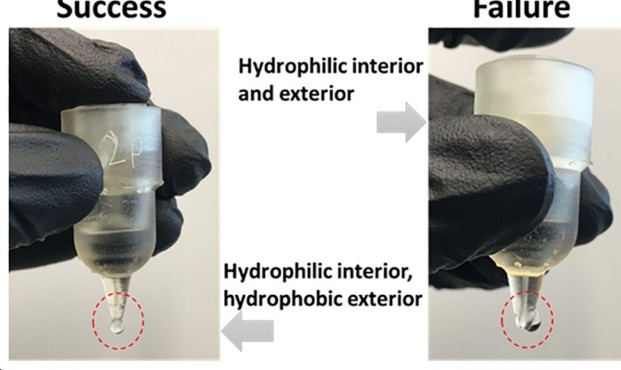

| | | No PEG 400 | PEG 400 (for 24 hr) | After 35 days of PEG 400 treatment (for 24 hr) |
|---|---|---|---|---|
| | No wrapped tip, No plasma | 16.8±1.72 | | |
| | No wrapped tip, With plasma (for 2 min) | 20.8±2.78 | 24.1±1.67 | |
| | Wrapped tip, With plasma (for 2 min) | 17.0±1.23 | 22.4±1.18 | 21.3±1.07 |
| | Wrapped tip, With plasma (for 4 min) | 19.6±1.86 | | |
| | Wrapped tip, After 10 days of plasma treatment (for 2 min) | 16.5±2.55 | | |

Capillary diameter = 0.7 mm
Capillary length = 11 mm

**Fig 2. Effects of various surface treatment procedures on the dispensing capacity of the drop dispenser.** a) Importance of having a hydrophilic interior and hydrophobic exterior surface for the liquid holder of the drop dispenser on the dispensing performance. b) Performance of drop dispensers treated by plasma and PEG 400 after a while of storage. All values are in µL.

the drop dispensers decreased over time (Fig 2B). Therefore, immediately after plasma treatment we also exposed the devices to PEG 400. The most satisfactory performance (i.e., less variability) was observed when the PEG 400 was poured within the liquid holder while the tip was wrapped with Parafilm wrapping film (Fig 2B). Such a treatment approach provided a liquid holder with a hydrophilic capillary tube and a hydrophobic outer tip surface. This feature helped the device to 1) hold the liquid within the capillary tube when the dispenser was not operating, 2) quickly refill the capillary tube after each dispense, and 3) prevent the drop from sticking to the tip outer surface during dispensing. After one week, we did not observe a notable change in the dispensing performance. Therefore, we proceeded with this surface treatment approach for additional devices.

Once we settled on the surface treatment approach, we fabricated and tested devices out of other resin types. A comparison among the results of drop dispensing for all other resins showed that the High Temp V2 resin was the best choice in terms of precision of drop volume and the least number of failed devices (Fig 3). Plasma and PEG 400 treatments were not effective on devices made from Rigid 4000 V1 resin and all devices failed due to uncontrolled dripping. Most of the devices made from BioMed Clear V1 resins had blocked capillary tubes after fabrication or surface treatment. Some of the devices made from Clear V4 resin provided similar dispensing performance as the High Temp V2 resin, however, most of them failed due to uncontrolled dripping (Fig 3).

Several geometric factors in the device were evaluated to determine their effects on the drop volumes [48]. Among these factors, the angle of the tip ($\alpha$) was the most effective (Fig 4, $\alpha$ angle of the tip; $\theta$ angle of the capillary channel). As shown in Fig 4, changing $\alpha$ from 0 to 15 degrees could change the capacity of the drop dispenser from 21.5 µL to 32.4 µL with reasonable precision. Increasing $\alpha$ to higher than 15 degrees led to uncontrolled dripping. All the upcoming experiments were performed using devices with $\alpha = 0$ since it helped to keep the LAMP reaction volume at 30 µL.

## 3.2 Precision of the drop dispensers over time of application

Six devices made out of High Temp V2 resin were used to evaluate their performance over time. The devices were stored at room temperature and used to dispense samples at various time intervals (0, 3, 7, 10, 14, 17, 21, 24, 28, and 35 days). The means of the measured drop volumes are plotted in Fig 5 and tabulated in S2 Table in S1 File. After 35 days of repeated use and devices stored at room temperature, some of the devices could no longer keep the liquid

**Effect of resin type on dispensing performance**

| Resin type | Device I | Device II | Device III | Device IV | Device V | Device VI |
|---|---|---|---|---|---|---|
| High Temp V2 | 22.4±1.18 | 21.8±1.00 | 22.2±0.69 | 21.9±1.23 | 20.9±0.84 | 21.6±0.96 |
| Clear V4 | 21.8±1.22 | Failed* | Failed | Failed | Failed | 22.1±1.39 |
| Rigid 4000 V1 | Failed | Failed | Failed | Failed | Failed | Failed |

* The device failed due to uncontrolled dripping.

Capillary diameter = 0.7 mm
Capillary length = 11 mm

**Fig 3. Effect of resin types on the performance of the drop dispensers.** All values are in µL.

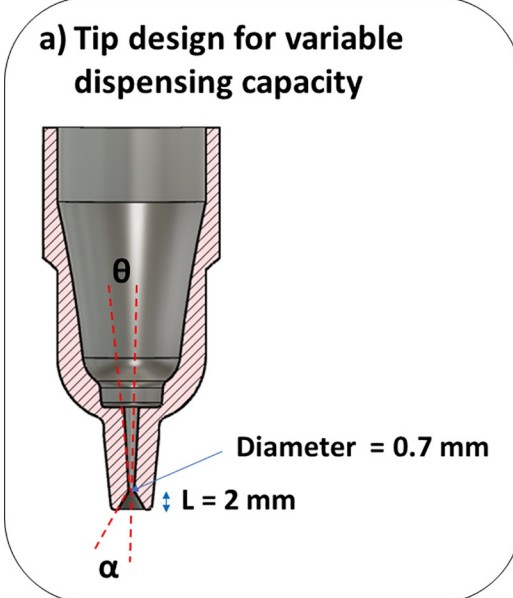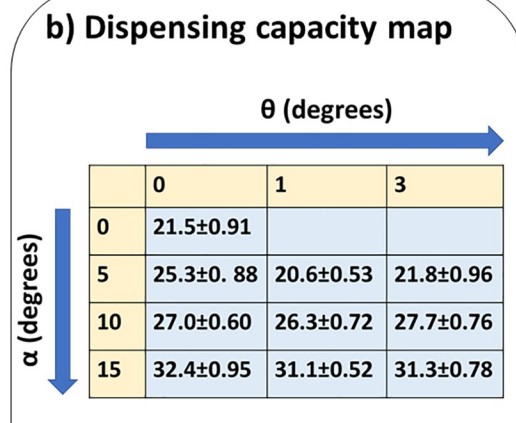

**Fig 4. Design of drop dispensers with variable volumes.** a) Description of conical tip design for generation of drops of different volumes. α angle of the tip; θ angle of the capillary channel b) Dispensing capacity for drop dispensers with variable volumes for different tip angles.

inside the liquid holder. Therefore, we considered the shelf life for the devices at room temperature to be 28 days. In order to understand how multiple devices and multiple days of repeated usage affect the drop dispensing performance, we conducted a two-way analysis of variance (ANOVA) on the obtained data set. As shown in Table 1, for a significant level of $\alpha = 0.001$, the effects of multiple devices, multiple days of application, and their interactions are significant. To further determine for which devices and on what days the performances were significantly different, we conducted a Tukey test to compare the means on various days. The results are shown in Table 2. Comparing the results corresponding to day 0 with day 28, two devices (i.e., I and V) showed significant differences in the drop volumes. For Device IV, no significant differences were observed among all days of usage. Also, two other devices (i.e., II and III) only showed significant differences among two or three compared cases. These observations highlight the potential of this design to generate statistically similar drop sizes for future applications. Some of this variability may be due to the use of resins for prototyping. For future fabrication in commercial use, these devices could be made using mold injection that can provide a better surface property, and therefore, a better drop dispensing performance, than resins. To further evaluate our devices, we tested some of the commercially available drop dispensers for repeated drop generation for multiple days. As shown in S1 Fig in S1 File, our design provides a comparable or better precision and reproducibility in drop generation than several commercial micro-pipettors that are meant for point-of-care applications. Since our device is eventually going to be used for on-farm LAMP assays, we investigated its performance for this purpose, as described below.

### 3.3 Efficacy of the drop dispenser in performing LAMP using extracted DNA

As discussed in the methods section "LAMP assays on extracted bacterial DNA using drop dispenser", we evaluated the performance of the drop dispenser in performing LAMP assays and

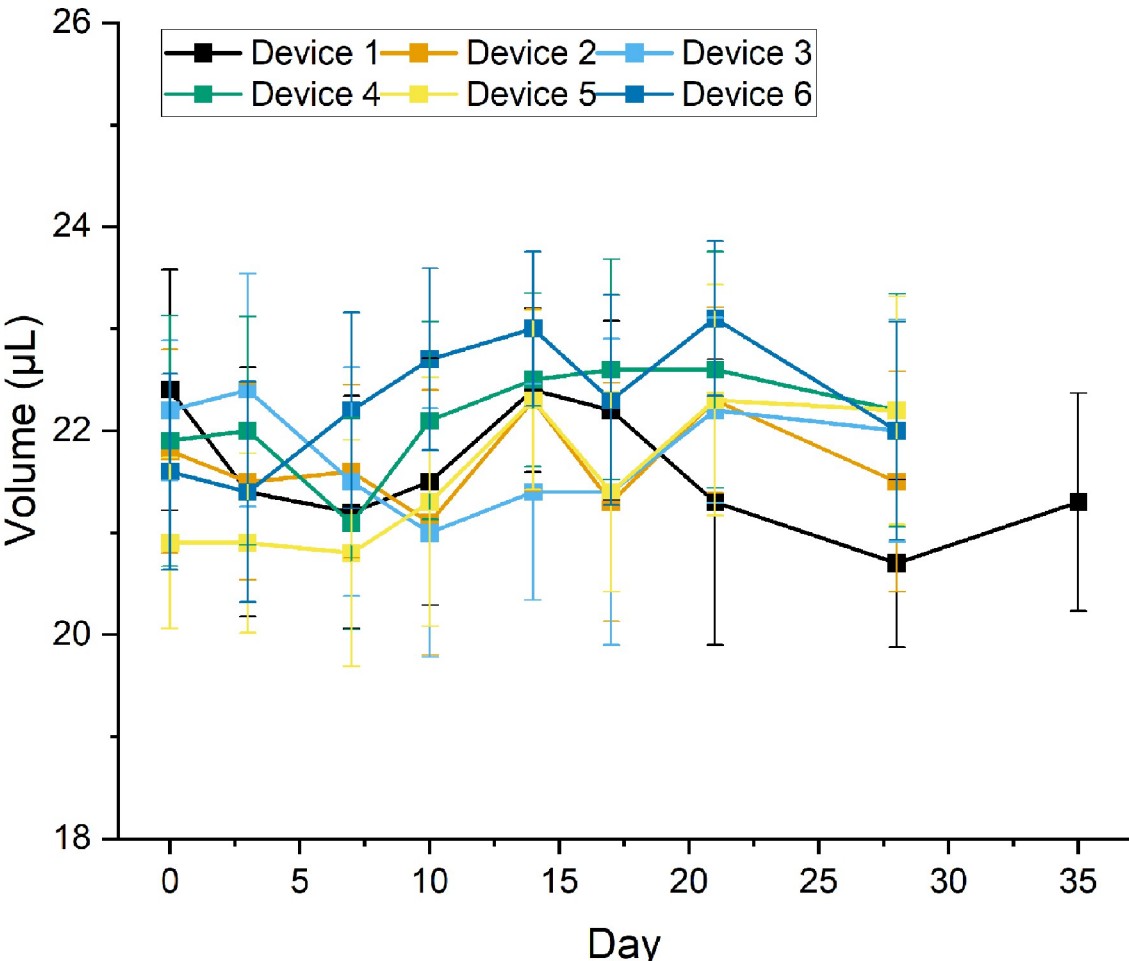

**Fig 5. Performance of the drop dispensers fabricated from High Temp V2 resin for several days after surface treatment with plasma and PEG 400.** Devices 2–6 failed on day 35 due to uncontrolled dripping after several uses.

compared the results to those of standard lab-based pipettors. Table 3 and S4-S6 Figs in S1 File show the representative LAMP results when using various concentrations of extracted DNA of *E. coli* O157:H7 as the template, and the EC.stx1.1 primer set (S1 Table in S1 File). In S4-S6 Fig in S1 File, a yellow color of the reaction mixes after 60 min of heating at 65˚C indicates a positive detection, while a red color indicates a negative detection. This experiment was repeated three times and in all of them, the LoD associated with the application of standard pipettors was consistently 5000 copies per reaction. When using the drop dispenser, however, we found an LoD of 2500 copies per reaction in two replicates (S4, S5 Figs in S1 File), and 1250 copies

**Table 1. Results of the two-way ANOVA on the performance data obtained from multiple testing of the drop dispenser.**

| Source of variation | SS | df | MS | F | P-value |
|---|---|---|---|---|---|
| Devices | 124.4395 | 5 | 24.88789 | 22.45187 | 1.04E-21 |
| Days of application | 113.1684 | 7 | 16.16692 | 16.16692 | 2E-18 |
| Interaction | 256.9596 | 35 | 7.341704 | 6.623098 | 4.14E-28 |
| Within | 1543.032 | 1392 | 1.1085 | | |
| Total | 2037.6 | 1439 | | | |

**Table 2. Results of the Tukey test on the performance of the drop dispensers over several days of application after PEG 400 treatment.**

| Compared days | Device I | Device II | Device III | Device IV | Device V | Device VI |
|---|---|---|---|---|---|---|
| 0–3 | – | – | – | – | – | – |
| 0–7 | * | – | – | – | – | – |
| 0–10 | – | – | * | – | – | – |
| 0–14 | – | – | – | – | * | * |
| 0–17 | – | – | – | – | – | – |
| 0–21 | * | – | – | – | * | * |
| 0–28 | * | – | – | – | * | – |
| 3–7 | – | – | – | – | – | – |
| 3–10 | – | – | * | – | – | * |
| 3–14 | – | – | – | – | * | * |
| 3–17 | – | – | – | – | – | – |
| 3–21 | – | – | – | – | * | * |
| 3–28 | – | – | – | – | * | – |
| 7–10 | – | – | – | – | – | – |
| 7–14 | * | – | – | – | * | – |
| 7–17 | – | – | – | – | – | – |
| 7–21 | – | – | – | – | * | – |
| 7–28 | – | – | – | – | * | – |
| 10–14 | – | * | – | – | – | – |
| 10–17 | – | – | – | – | – | – |
| 10–21 | – | * | * | – | – | – |
| 10–28 | – | – | – | – | – | – |
| 14–17 | – | – | – | – | – | – |
| 14–21 | * | – | – | – | – | – |
| 14–28 | * | – | – | – | – | – |
| 17–21 | – | – | – | – | – | – |
| 17–28 | * | – | – | – | – | – |
| 21–28 | – | – | – | – | – | – |

* The absolute difference of the means was calculated from the ANOVA results. Critical value = Q *sqrt(MS/n.obs), Q: from Studentized Range q Table, MS: From two-way ANOVA results, n.obs: number of observation for each test = 30. The critical value at alpha = 0.001 was 1.11931935.–means there is no significant difference; * means there is a significant difference.

per reaction in another replicate (S6 Fig in S1 File). It's unclear to us why the drop dispenser provided a better LoD as compared to a standard pipettor, however, our conjecture is that it might be due to a lower adhesion of the DNA in the liquid holder of the drop dispenser (as compared to a pipette tip) before adding it to the LAMP tubes. There were no false positives in all replicates. These results indicate that the drop dispenser can be successfully applied in the detection of bacterial DNA. Since the final goal is to use this device for whole-cell LAMP, we conducted a number of experiments to examine this capability, as elaborated below.

### 3.4 Efficacy of the drop dispenser on performing LAMP using bacteria swabbed from collection flags

Our previous work [6] confirmed that the use of collection flags—made from clean plastic sheets—instead of lettuce leaves, provided more repeatable results for bioaerosol sample collection in the field. This was because, unlike the lettuce leaves, collection flags were flat and did not include various microstructures on their surfaces. Therefore, their swabbing was much

**Table 3. Results of the LAMP assays using the drop dispenser vs. a standard pipettor.**

| DNA concentration (copies/reaction) | Device Assay 1 | Device Assay 2 | Device Assay 3 | Pipettor Assay 1 | Pipettor Assay 2 | Pipettor Assay 3 |
|---:|:---:|:---:|:---:|:---:|:---:|:---:|
| 10,000 | ■ ■ ■ | ■ ■ ■ | ■ ■ ■ | ■ ■ ■ | ■ ■ ■ | ■ ■ ■ |
| 5000 | ■ ■ ■ | ■ ■ ■ | ■ ■ ■ | ■ ■ ■ | ■ ■ ■ | ■ ■ ■ |
| 2500 | ■ ■ ■ | ■ ■ ■ | ■ ■ ■ | ■ ■ ■ | □ □ ■ | ■ □ ■ |
| 1250 | □ □ □ | ■ ■ □ | ■ □ □ | ■ □ □ | □ ■ ■ | ■ □ ■ |
| 500 | □ □ □ | □ □ □ | ■ □ □ | □ □ □ | ■ ■ □ | □ □ □ |
| 250 | □ □ ■ | □ □ □ | □ □ □ | □ □ □ | □ □ □ | □ ■ □ |
| 0 | □ □ □ | □ □ □ | □ □ □ | □ □ □ | □ □ □ | □ □ □ |

Boxes that are empty (not filled) indicate that there was no color change in the reaction mix after the addition of the DNA template and heating for 1 hr at 65˚C; Boxes that are filled indicate that there was a color change in the reaction mix, from red to yellow, after adding the DNA template and heating for 1 hr at 65˚C.

more efficient in the collection of the microbial and DNA samples from their surfaces. As discussed in the methods section for "LAMP assays on samples from artificially contaminated collection flags using drop dispenser," we manually contaminated a number of collection flags with various dilutions of *E. coli* O157:H7 culture and used the swabbed and resuspended cells for LAMP assays. This approach was a simulation of on-farm assays that our drop dispenser is being developed for.

In general, these assays showed that the performance of the drop dispenser is similar to that of the pipettor (Table 4 and S7 Fig in S1 File). Here, the LoD for bacterial detection was 7.8 $\times10^6$ cells/mL (i.e., $1.6\times10^5$ DNA copies/reaction). The process of swabbing for microbial collection from artificially contaminated plastic sheets may lose some of the cells when compared with the direct addition of the bacteria from a culture. In order to investigate this issue, we ran a separate LoD test using bacterial cells (from the same dilutions) directly added to the reaction sites. However, as shown in S8 Fig in S1 File, the results of this assay were similar to those when swabbing the flags, with a similar LoD of $7.8\times10^6$ cells/mL (i.e., $1.6\times10^5$ DNA copies/reaction). This highlights the performance of the swabbing technique for on-farm microbial collection using collection flags. A comparison of the results in Table 4 with those in Table 3 shows that the LoD for LAMP using whole-cell samples was about two orders of magnitude worse than when using extracted DNA samples. This might be because, in whole-cell LAMP assays, the bacterial genome is not readily available for amplification until the cells are lysed and their membranes are ruptured during the heating process.

**Table 4. Results of the whole-cell LAMP assays using the drop dispenser vs. a standard pipettor.**

| Swabbed samples (dilution factor) | Device | Pipettor |
|:---|:---:|:---:|
| Bacteria[a] (1) | ■ ■ ■ | ■ ■ ■ |
| Bacteria (10) | ■ ■ ■ | ■ ■ ■ |
| Bacteria (100) | ■ ■ ■ | ■ ■ ■ |
| Bacteria (1000) | ■ ■ □ | □ ■ □ |
| Bacteria (10000) | □ □ ■ | □ □ □ |
| Water | □ □ □ | □ □ □ |
| Bacterial DNA extract[b] | ■ ■ ■ | ■ ■ ■ |

\* Boxes that are empty (not filled) indicate that there was no color change in the reaction mix after the addition of the swabbed sample and heating for 1 hr at 65˚C; Boxes that are filled indicate that there was a color change in the reaction mix, from red to yellow, after the addition of the swabbed sample and heating for 1 hr at 65˚C

[a]The initial concentration of the bacteria, Shiga-toxin-producing *E. coli* O157 (ATCC 8739) was $7.8\times10^8$ cells/mL

[b]The concentration of bacterial DNA extract was $1\times10^5$ copies/µL.

## 4. Discussion

A number of commercial portable instruments to perform LAMP at the point of care (POC) are available. Examples of these devices include the 3M Molecular Detection System (3M Science. Applied to Life., MN) for the detection of food-borne pathogens such as *Salmonella spp.* and *Listeria monocytogenes*, HumaLoop T and HumaLoop M (HUMAN Diagnostics, Germany) for detection of *Mycobacterium tuberculosis* complex, *Plasmodium spp.*, Genie II (Opti-Gene Ltd., UK) for detection of plant pathogens, and CapitalBio RTisochip (CapitalBio Technology, China) for applications such as food safety testing, clinical diagnosis, etc. The majority of the available tools are challenging to use in the field by non-specialist personnel and require multiple handling of the reagents using pipettors. If the LAMP POC device was fast and straightforward to use, it could be adopted by industrial stakeholders for decision-making during pre-season or pre/post-harvest of fresh produce [49, 50]. Within the pre-season period, these tools could be used for risk assessment of pathogen contamination from nearby animal operations [6]. For pre-harvest, the tool could help decide whether the harvested produce is safe to be sent to the market [51, 52]. Therefore, it is essential to decrease the preparation steps of the future fully integrated biosensors that are meant to be used in the field.

The drop dispenser introduced in this work is meant to satisfy this goal. As compared with the evaluated commercial pipettors, this device provides high precision in dispensing a fixed drop volume (S1 Fig in S1 File). Low-end devices such as capillary micropipettes use sharp glass capillary tubes. Because both the interior and exterior surfaces of the glass capillary tubes are hydrophilic, there is always a chance of the drop remaining stuck to the exhaust tip of the capillary tube; reducing the dispensing precision, which is necessary for getting a reliable and consistent result for nucleic acid assays. Furthermore, when not handled or disposed of properly, glass sharps can be potentially hazardous for on-farm applications by non-specialist users. Other types of low-end sample delivery devices such as squeezing plastic tubes also do not provide high-precision drop volumes (S1 Fig in S1 File). These devices are commonly used in commercial lateral flow assays such as the CareStart COVID-19 Antigen Home Test (Access Bio, CA), and the BD Veritor At-Home COVID-19 Test (Becton Dickinson, NJ), where consistency of sample volume is not critical. While high-end devices like lab pipettes are very accurate for liquid handling, their cost can be a limiting factor. Additionally, it is crucial to consider that when used by non-specialist users, improper handling or misuse can lead to cross-contamination between samples. This concern becomes particularly critical for on-site assays where the protocols and controlled experimental environment typically found in laboratory settings may not be feasible. Therefore, in the context of user-friendly on-farm applications, it is essential to address the challenges with these currently available liquid dispensers and provide solutions that mitigate the risks of cross-contamination while still maintaining accuracy in sample handling.

Our device takes advantage of a specific surface chemistry treatment to create a hydrophilic interior surface of the capillary tube and a hydrophobic exterior surface of the tip, enhancing the dispensing precision. Utilizing a positive displacement dispensing mechanism, we ensure precise and accurate delivery of the desired sample volume. Additionally, our device is disposable, using one dispenser for each sample, which will minimize the possibility of cross-contamination.

Furthermore, most commercially available devices are standalone designs, making it difficult to integrate them into fully integrated sample-to-answer nucleic-acid biosensors. With this custom-designed device, we will be able to add additional functionalities in the future, such as sample preparation and multichannel fluid dispensing. These advancements facilitate streamlined workflow, improve user convenience, and reduce the possibility of handling errors

or contamination. These improvements set our device apart, offering reliable and consistent results for the testing of nucleic acids. We envision our drop dispenser as an essential part of a future portable LAMP biosensor to run easily and properly in the field. Based on our estimation (S3 Table in S1 File), running LAMP using this device will cost less than USD 4 which is reasonable for future industrial applications. More than 50% of the total price corresponds to the resin cost that can be decreased when the device is mass-fabricated from plastics using injection molding.

Our drop dispenser will eventually be used as part of a kit with a paper-based LAMP biosensor [43]. The papers will be placed inside a cartridge which is coupled with the drop dispenser as a disposable unit which would cost about USD 7 per test, based on the paper-based LAMP cost estimation provided by Davidson *et al.* in addition to the fabrication costs for the drop dispenser in S3 Table in S1 File. The user will only add the DNA/RNA sample to the liquid holder and press the plunger to deliver the sample to the paper pads within the cartridge that hold all the needed reagents in a dry state [53]. Currently, such an easy-to-use device for on-farm applications does not exist commercially. So far, our efforts have been directed toward enabling LAMP assays for on-farm applications [4, 5] and the presented drop dispenser is another step forward to achieving this goal.

## 5. Conclusions

We have developed a drop dispenser for nucleic-acid testing to be used as part of a biosensor for on-farm microbial risk assessments. This device offers the following four advantages: i) it provides precise and reproducible drops for downstream nucleic-acid testing. ii) it employs specialized surface chemistry treatment and a positive displacement mechanism for accurate sample delivery. iii) it is disposable to minimize cross-contamination. iv) the simple design is suitable for high-volume manufacturing. Our device also provided similar or lower limits of detection in LAMP assays on bacterial cells or DNA, when compared with standard laboratory pipettors.

However, the current approach has three limitations: i) the device's performance is highly dependent on specific resin types and surface chemistry treatment. ii) it requires a rigorous washing protocol for optimal performance. iii) the range of drop volume, while precise, is limited to 20 to 33 μL.

Our drop dispenser will be part of a consumable kit for a paper-based LAMP biosensor which will be used for on-farm risk assessment of fresh produce. Given the process of fluid delivery operates manually, the cost of the final consumable kit is affordable (around USD 4 per test). More investigation is needed to test this drop dispenser on-farm using microbial samples collected from produce leaves/tissue or collection flags distributed on the farm. Also, coupling the drop dispensers with paper-based LAMP biosensors to form a consumable kit is another area that needs further research [54, 55].

## Supporting information

**S1 File.**
(DOCX)

## Acknowledgments

The plasma treatments were performed in Prof. Rahim Rahimi's laboratory located at the Birck Nanotechnology Center of Purdue University. Sina Nejati, his graduate student, provided initial training to use the plasma generator. Andres A. Dextre, an undergraduate

researcher at Prof. Mohit Verma's group helped in setting reagent concentrations used in the LAMP master mix.

## Author Contributions

**Conceptualization:** Mohsen Ranjbaran, Jiangshan Wang, Mohit S. Verma.

**Data curation:** Mohsen Ranjbaran.

**Funding acquisition:** Mohit S. Verma.

**Investigation:** Mohsen Ranjbaran, Simerdeep Kaur, Jiangshan Wang.

**Project administration:** Mohit S. Verma.

**Supervision:** Mohit S. Verma.

**Visualization:** Mohsen Ranjbaran.

**Writing – original draft:** Mohsen Ranjbaran.

**Writing – review & editing:** Mohsen Ranjbaran, Simerdeep Kaur, Jiangshan Wang, Bibek Raut, Mohit S. Verma.

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
