## [Decision Letter · Decision Letter 0]

22 Aug 2024

PONE-D-24-22915A drop dispenser for simplifying on-farm detection of foodborne pathogensPLOS ONE

Dear Dr. Verma,

Thank you for submitting your manuscript to PLOS ONE. After careful consideration, we feel that it has merit but does not fully meet PLOS ONE’s publication criteria as it currently stands. Therefore, we invite you to submit a revised version of the manuscript that addresses the points raised during the review process.

Please submit your revised manuscript by Oct 06 2024 11:59PM. If you will need more time than this to complete your revisions, please reply to this message or contact the journal office at plosone@plos.org. Please include the following items when submitting your revised manuscript:A rebuttal letter that responds to each point raised by the academic editor and reviewer(s). You should upload this letter as a separate file labeled 'Response to Reviewers'.A marked-up copy of your manuscript that highlights changes made to the original version. You should upload this as a separate file labeled 'Revised Manuscript with Track Changes'.An unmarked version of your revised paper without tracked changes. You should upload this as a separate file labeled 'Manuscript'.

We look forward to receiving your revised manuscript.

Kind regards,

Enoch Aninagyei, PhD

Academic Editor

PLOS ONE

Journal Requirements:

"The work presented here is funded by CPS AWARD NUMBER: 2021CPS12, CDF Agreement No: 20-0001-054-SF USDA Cooperative Agreement No. USDA-AMS-TM-SCBGP-G-20-0003. Any opinions, findings, conclusions, or recommendations expressed in this publication or audiovisual are those of the author(s) and do not necessarily reflect the views of The Center for Produce Safety, the California Department of Food and Agriculture, or the Agricultural Marketing Service of the U.S. Department of Agriculture (USDA). The work upon which this project entitled “Field evaluation of microfluidic paper-based analytical devices for microbial source tracking” was funded in whole or in part through a subrecipient grant awarded to The Center for Produce Safety through the California Department of Food and Agriculture 2020 Specialty Crop Block Grant Program and the USDA’s Agricultural Marketing Service."

"I have read the journal's policy and the authors of this manuscript have the following competing interests: M.S.V. has an interest in Krishi Inc., which is a startup that is interested in commercializing technologies developed here. This work was not funded by Krishi Inc. "

4. We noted in your submission details that a portion of your manuscript may have been presented or published elsewhere. [The manuscript has been deposited to bioRxiv as a pre-print.] Please clarify whether this [conference proceeding or publication] was peer-reviewed and formally published. If this work was previously peer-reviewed and published, in the cover letter please provide the reason that this work does not constitute dual publication and should be included in the current manuscript.

Reviewers' comments:

Reviewer's Responses to Questions

**Comments to the Author**

1. Is the manuscript technically sound, and do the data support the conclusions?

Reviewer #1: Yes

Reviewer #2: Yes

Reviewer #3: Yes

2. Has the statistical analysis been performed appropriately and rigorously? 

Reviewer #1: Yes

Reviewer #2: Yes

Reviewer #3: Yes

3. Have the authors made all data underlying the findings in their manuscript fully available?

Reviewer #1: Yes

Reviewer #2: Yes

Reviewer #3: Yes

4. Is the manuscript presented in an intelligible fashion and written in standard English?

Reviewer #1: Yes

Reviewer #2: Yes

Reviewer #3: Yes

5. Review Comments to the Author

Reviewer #1: The manuscript titled, “A drop dispenser for simplifying on-farm detection of foodborne pathogens” discusses the fabrication of drop dispensers for efficient and cost-effective nucleic acid testing while delivering precise sample amounts to biosensor’s reaction sites for detecting food-borne pathogens. This study utilizes loop-mediated isothermal amplification (LAMP) to detect Ecoli. The introduction of the manuscript makes a strong case for the need and applicability of the method. The results demonstrate a proof of concept in terms of use in the field by non-specialists. The one limitation is that the manuscript does not discuss how the dispenser performs over time with repeated usage. Are desired volumes dispensed by the method still accurate over time? This should be addressed in the comments. I recommend the manuscript for publication once this minor comment is addressed.

Reviewer #2: The manuscript entitled A drop dispenser for simplifying on-farm detection of foodborne pathogens by Mohsen Ranjbaran in which the authors aimed to developed drop dispensers using 3D printing and a hydrophilic surface chemistry treatment to generate precise drops (DNA/bacterial samples) of a few micro-liters (∼20 to ∼33 μL).

The idea of the manuscript is good and can be accepted for publication but after some revision.

The authors should revise their manuscript according to the following comments.

Abstract

rewrite it and make it more coherent.

Introduction

Should be revised and updated.

Latest literature should be added, recent references should be cited.

Discussion

The authors have mentioned listeria but they havent mentioned more detail about it.

As listeria is a food born pathogen and very harmful, the authors should add at least a paragraph on it.

The following two articles are the best to be read and cited for writing the new paragraph.

https://doi.org/10.1080/19476337.2023.2296006

https://doi.org/10.31083/j.fbl2905176.

Figures quality should be improved.

Where is the conclusion of the study?

Limitation and future recommendation should also be added as a separate paragraph.

Reviewer #3: The current study outlines the development of a drop dispenser for application in point-of-care detection of food borne pathogens in the field. Overall, the report is thorough and well-written, with plenty of optimization done to develop a robust device and thoughtful discussion of the state of the field. This study and device would be very helpful to others in the area.

Major Notes

- There needs to be some more discussion of how well the device would function in the field. Further, are there any obvious limitations and potential future steps to improve them?

- For instance, currently the dispenser works to detect 8e6 bacterial cells per mL - is this a reasonable LOD for POC E Coli detection? If not, how could the device be improved to get there

- I have also highlighted several ways to potentially improve the data presentation by moving numbers from tables to visually comprehensible plots and heat maps. Also, since the report currently only has two figures in the main text, I’d suggest moving some key supplementary figures (Fig S4 and 5 for eg) to the main text since they include important results that supported the development of the device.

Minor notes

- Please order figures and supplemental figures in the order that they are cited in the text. Currently, Figure S1 for example is cited after Figures S4 and S5

- Table 1: Currently the text makes it seems like the devices were storing liquid over 35 days with some dropping at the indicated time points. Would the devices ever realistically store field samples for that length of time in the POC setting?

- If so, are there potential improvements that can be made to the device in the future to limit the liquid leaking issue after day 28?

- If not, what’s a more reasonable storage timeline in the field? Please add to discussion

- Table 1: I’d suggest converting this table to a figure to make it easier to interpret visually

- Also, how many drops were analyzed to determine the error? Please include either individual markers if less than five or error bars to demonstrate the spread of results graphically

- In general, please add the numbers of drops analyzed to determine the error in all figure and table captions along with the methods

- I’d also suggest converting Tables 4 and 5 to heatmaps for better visual presentation

- Figure 2: the text states that various factors were tested but we only see data for alpha and theta (angle of the capillary tube?) with no in-text discussion for theta. Please add a statement defining what theta is and what impact even if none on the drop volume. Further, either include data for volume of the dispensing cylinder, diameter of the capillary tube etc. or don’t mention them as factors tested at all

- Figure S3: please plot this numbers instead of listing them

- Figure S6a-c: all captions say “replicate 1”. Are they supposed to say replicate 2 or 3?

- Where were the primer sequences for the LAMP reaction obtained from? And what vendor did the authors use to synthesize the primers? Please include in Methods

- Discussion line 410-411: here the authors state that “Based on our results, our devices worked better than the available commercial drop dispensers in terms of drop reproducibility and application simplicity.” If the authors are referring to results in Figure S1, they need to perform statistical significance testing to conclusively state of their device outperformed the others tested. Further, since “application simplicity” is a subjective metric, I would not include it here and rely on other text in the Discussion to make that point.

6. PLOS authors have the option to publish the peer review history of their article (what does this mean?). If published, this will include your full peer review and any attached files.

Reviewer #1: No

Reviewer #2: No

Reviewer #3: No

---

## [Author Response · Author response to Decision Letter 0]

9 Oct 2024

Editor and Reviewers’ Comments and Authors’ Responses

Editor’s comments:

 Please ensure that your manuscript meets PLOS ONE's style requirements, including those for file naming. The PLOS ONE style templates can be found at https://journals.plos.org/plosone/s/file?id=wjVg/PLOSOne_formatting_sample_main_body.pdf and https://journals.plos.org/plosone/s/file?id=ba62/PLOSOne_formatting_sample_title_authors_affiliations.pdf

Author’s response: 

We formatted the manuscript to meet PLOS ONE’s style requirements.

Author’s response: 

We added the requested statement at the end of the funding section. 

“The work presented here is funded by CPS AWARD NUMBER: 2021CPS12, CDF Agreement No: 20-0001-054-SF USDA Cooperative Agreement No. USDA-AMS-TM-SCBGP-G-20-0003. Any opinions, findings, conclusions, or recommendations expressed in this publication or audiovisual are those of the author(s) and do not necessarily reflect the views of The Center for Produce Safety (CPS), the California Department of Food and Agriculture, or the Agricultural Marketing Service of the U.S. Department of Agriculture (USDA). The work upon which this project entitled “Field evaluation of microfluidic paper-based analytical devices for microbial source tracking” was funded in whole or in part through a subrecipient grant awarded to CPS through the California Department of Food and Agriculture 2020 Specialty Crop Block Grant Program and the USDA’s Agricultural Marketing Service. The funders had no role in study design, data collection and analysis, decision to publish, or preparation of the manuscript. The funding organization (CPS) reviewed the manuscript prior to submission to ensure the accuracy of the information and address any concerns about the publication of sensitive content." 

 Please confirm that this does not alter your adherence to all PLOS ONE policies on sharing data and materials, by including the following statement: "This does not alter our adherence to PLOS ONE policies on sharing data and materials.” (as detailed online in our guide for authors http://journals.plos.org/plosone/s/competing-interests). If there are restrictions on sharing of data and/or materials, please state these. Please note that we cannot proceed with consideration of your article until this information has been declared. Please include your updated Competing Interests statement in your cover letter; we will change the online submission form on your behalf.

Author’s response: 

We confirm that this does not alter the adherence to all PLOS ONE policies on sharing data and materials and added the statement “This conflict does not alter our adherence to PLOS ONE policies on sharing data and materials.” in the “Conflict of Interest” section. 

 We noted in your submission details that a portion of your manuscript may have been presented or published elsewhere. [The manuscript has been deposited to bioRxiv as a pre-print.] Please clarify whether this [conference proceeding or publication] was peer-reviewed and formally published. If this work was previously peer-reviewed and published, in the cover letter please provide the reason that this work does not constitute dual publication and should be included in the current manuscript. 

Author’s response: 

This manuscript has not been peer-reviewed or published elsewhere. BioRxiv is an online archive and distribution service for unpublished preprints. 

 Please include captions for your Supporting Information files at the end of your manuscript, and update any in-text citations to match accordingly. Please see our Supporting Information guidelines for more information: http://journals.plos.org/plosone/s/supporting-information. 

Author’s response: 

We included captions for the Supporting Information files at the end of the manuscript and updated in-text citations to match accordingly.

Reviewers' comments:

Reviewer #1

The manuscript titled, “A drop dispenser for simplifying on-farm detection of foodborne pathogens” discusses the fabrication of drop dispensers for efficient and cost-effective nucleic acid testing while delivering precise sample amounts to biosensor’s reaction sites for detecting food-borne pathogens. This study utilizes loop-mediated isothermal amplification (LAMP) to detect Ecoli. The introduction of the manuscript makes a strong case for the need and applicability of the method. The results demonstrate a proof of concept in terms of use in the field by non-specialists. The one limitation is that the manuscript does not discuss how the dispenser performs over time with repeated usage. Are desired volumes dispensed by the method still accurate over time? This should be addressed in the comments. I recommend the manuscript for publication once this minor comment is addressed.

 The manuscript does not discuss how the dispenser performs over time with repeated usage. Are desired volumes dispensed by the method still accurate over time?

Author’s response:

We discussed how the dispensers perform over time with repeated usage in section 3.2. In this section, we concluded that the shelf life for the devices at room temperature to be 28 days. In addition, to assess which devices and which days showed significant differences in performance, we used a Tukey test to compare the means on different days. Table 2 shows that, while some devices showed variability (alpha < 0.001) at different time points during the 28-day research, most of the devices maintained accuracy throughout the testing period.

Reviewer #2

The manuscript entitled A drop dispenser for simplifying on-farm detection of foodborne pathogens by Mohsen Ranjbaran in which the authors aimed to developed drop dispensers using 3D printing and a hydrophilic surface chemistry treatment to generate precise drops (DNA/bacterial samples) of a few micro-liters (∼20 to ∼33 μL). The idea of the manuscript is good and can be accepted for publication but after some revision. The authors should revise their manuscript according to the following comments.

 Abstract: rewrite it and make it more coherent. 

Author’s response:

We have rewritten the Abstract.

“Nucleic-acid biosensors have emerged as useful tools for on-farm detection of foodborne pathogens on fresh produce. Such tools are specifically designed to be user-friendly so that a producer can operate them with minimal training and in a few simple steps. However, one challenge in the deployment of these biosensors is delivering precise sample volumes to the biosensor's reaction sites. To address this challenge, we developed an innovative drop dispenser using advanced 3D printing technology, combined with a hydrophilic surface chemistry treatment. This dispenser enables the generation of precise sample drops, containing DNA or bacterial samples, in volumes as small as a few micro-liters (∼20 to ∼33 µL). The drop generator was tested over an extended period to assess its durability and usability over time. The results indicated that the drop dispensers have a shelf life of approximately one month. In addition, the device was rigorously validated for nucleic acid testing, specifically by using loop-mediated isothermal amplification (LAMP) for the detection of Escherichia coli O157, a prevalent foodborne pathogen. To simulate real-world conditions, we tested the drop dispensers by integrating them into an on-farm sample collection system, ensuring they deliver samples accurately and consistently for nucleic acid testing in the field. Our results demonstrated similar performance to commercial pipettors in LAMP assays, with a limit of detection of 7.8×10^6 cells/mL for whole-cell detection. This combination of precision, ease of use, and durability make our drop dispenser a promising tool for enhancing the effectiveness of nucleic acid biosensors in the field.”

 Introduction: Should be revised and updated. Latest literature should be added, recent references should be cited. 

Author’s response:

We have updated the Introduction section and added the following references.

 Carole, N.V.D., Sheng, L., Ji, J., Wu, S., Zhang, Y., Sun, X., 2025. Food Control 167, 110774.

 Li, Z., Bai, Y., You, M., Hu, J., Yao, C., Cao, L., Xu, F., 2021. Biosensors and Bioelectronics 177, 112952.

 Wang, F., Jiang, L., Ge, B., 2012. Journal of Clinical Microbiology 50, 91–97.

 Wang, J., Kaur, S., Kayabasi, A., Ranjbaran, M., Rath, I., Benschikovski, I., Raut, B., Ra, K., Rafiq, N., Verma, M.S., 2024. Biosensors and Bioelectronics 259, 116374.

 Xie, M., Chen, T., Xin, X., Cai, Z., Dong, C., Lei, B., 2022. Food Control 136, 108824.

 Discussion: The authors have mentioned listeria but they havent mentioned more detail about it. As listeria is a food born pathogen and very harmful, the authors should add at least a paragraph on it. The following two articles are the best to be read and cited for writing the new paragraph. https://doi.org/10.1080/19476337.2023.2296006
https://doi.org/10.31083/j.fbl2905176. 

Author’s response:

Thank you for your comments. We concur that Listeria is a severe foodborne pathogen that poses a significant threat to public health. However, this was not the reason we mentioned Listeria or Salmonella in the text. We wanted to compare the device presented in this study to commercial portable devices in the market. One of which is capable of identifying Listeria and Salmonella.

The articles that are suggested by the reviewer to review and cite focus on the development of multi-epitope vaccines against Listeria monocytogenes. These approaches for developing vaccines are a valuable tool against foodborne pathogens. However, the current manuscript focuses on detection tools and thus, mentioning details about Listeria or other foodborne pathogens is beyond the scope of this manuscript. 

 Figures quality should be improved. 

Author’s response:

In the manuscript submission, we have now included the high-quality .tif files for the figures individually. 

 Where is the conclusion of the study? 

Author’s response:

We separated the Conclusion from the Discussion section and made it an independent section. We revised the Conclusion section to improve clarity and readability.

“We have developed a drop dispenser for nucleic-acid testing to be used as part of a biosensor for on-farm microbial risk assessments. This device offers the following four advantages: i) it provides precise and reproducible drops for downstream nucleic-acid testing. ii) it employs specialized surface chemistry treatment and a positive displacement mechanism for accurate sample delivery. iii) it is disposable to minimize cross-contamination. iv) the simple design is suitable for high-volume manufacturing. Our device also provided similar or lower limits of detection in LAMP assays on bacterial cells or DNA, when compared with standard laboratory pipettors.

However, the current approach has three limitations: i) the device's performance is highly dependent on specific resin types and surface chemistry treatment. ii) it requires a rigorous washing protocol for optimal performance. iii) the range of drop volume, while precise, is limited to 20 to 33 µL.

Our drop dispenser will be part of a consumable kit for a paper-based LAMP biosensor which will be used for on-farm risk assessment of fresh produce. Given the process of fluid delivery operates manually, the cost of the final consumable kit is affordable (around USD 4 per test). More investigation is needed to test this drop dispenser on-farm using microbial samples collected from produce leaves/tissue or collection flags distributed on the farm. Also, coupling the drop dispensers with paper-based LAMP biosensors to form a consumable kit is another area that needs further research.(Ongaro et al., 2022; Yetisen et al., 2013) ”

 Limitation and future recommendation should also be added as a separate paragraph.

Author’s response:

We separated the limitation and future recommendation into a separate paragraph.

Reviewer #3:

The current study outlines the development of a drop dispenser for application in point-of-care detection of food borne pathogens in the field. Overall, the report is thorough and well-written, with plenty of optimization done to develop a robust device and thoughtful discussion of the state of the field. This study and device would be very helpful to others in the area.

 There needs to be some more discussion of how well the device would function in the field. Further, are there any obvious limitations and potential future steps to improve them? For instance, currently the dispenser works to detect 8e6 bacterial cells per mL - is this a reasonable LOD for POC E Coli detection? If not, how could the device be improved to get there 

Author’s response:

No, this is not a reasonable LoD for E coli. detection. However, a sample distribution system alone cannot resolve this situation. Improvement of the LoD will require optimization of the assay and sample processing. For instance, with heat lysis prior to sample addition, the LoD can be improved to 1000 copies/reaction (Kaur et al., 2024). Since this manuscript focuses on hardware, we did not emphasize assay tuning and instead compared it to other commercially available liquid dispensing devices. 

 I have also highlighted several ways to potentially improve the data presentation by moving numbers from tables to visually comprehensible plots and heat maps. Also, since the report currently only has two figures in the main text, I’d suggest moving some key supplementary figures (Fig S4 and 5 for eg) to the main text since they include important results that supported the development of the device.

Author’s response:

Thank you for your comment. We have converted Table 1, Table 4, and Table 5 into graphs and heat maps. We moved Fig S4 and 5 to the main text (now Fig 3 and 4). 

 Please order figures and supplemental figures in the order that they are cited in the text. Currently, Figure S1 for example is cited after Figures S4 and S5 

Author’s response:

We confirmed that all other figures and tables were cited in order. 

 Table 1: Currently the text makes it seems like the devices were storing liquid over 35 days with some dropping at the indicated time points. Would the devices ever realistically store field samples for that length of time in the POC setting? If so, are there potential improvements that can be made to the device in the future to limit the liquid leaking issue after day 28? If not, what’s a more reasonable storage timeline in the field? Please add to discussion 

Author’s response:

The experiments were designed to show that the devices (without any liquid) can last 35 days at room temperature. Over time, the PEG treatment will lose effectiveness. There is no point in storing the samples inside the device for such an extended period of time since it is meant to be tested right away. We improved the language to make it clearer.

“Six devices made out of High Temp V2 resin were used to evaluate their performance over time. The devices were stored at room temperature and used to dispense samples at various time intervals (0, 3, 7, 10, 14, 17, 21, 24, 28, and 35 days). The means of the measured drop volumes are tabulated in Table 1. After 35 days of repeated use and devices stored at room temperature, some of the devices could no longer keep the liquid inside the liquid holder.”

 Table 1: I’d suggest converting this table to a figure to make it easier to interpret visually 

Author’s response:

We converted Table 1 to a new figure and added the original Table 1 as S2 Table in SI.

Fig 5. Performance of the drop dispensers fabricated from High Temp V2 resin for several days after surface treatment with plasma and PEG 400. Devices 2-6 failed on day 35 due to uncontrolled dripping after several uses. For

---

## [Decision Letter · Decision Letter 1]

26 Nov 2024

A drop dispenser for simplifying on-farm detection of foodborne pathogens

PONE-D-24-22915R1

Dear Dr. Mohit Verma,

We’re pleased to inform you that your manuscript has been judged scientifically suitable for publication and will be formally accepted for publication once it meets all outstanding technical requirements.

Kind regards,

Enoch Aninagyei, PhD

Academic Editor

PLOS ONE

Additional Editor Comments (optional):

Reviewers' comments:

Reviewer's Responses to Questions

**Comments to the Author**

1. If the authors have adequately addressed your comments raised in a previous round of review and you feel that this manuscript is now acceptable for publication, you may indicate that here to bypass the “Comments to the Author” section, enter your conflict of interest statement in the “Confidential to Editor” section, and submit your "Accept" recommendation.

Reviewer #4: All comments have been addressed

2. Is the manuscript technically sound, and do the data support the conclusions?

Reviewer #4: Yes

3. Has the statistical analysis been performed appropriately and rigorously? 

Reviewer #4: Yes

4. Have the authors made all data underlying the findings in their manuscript fully available?

Reviewer #4: Yes

5. Is the manuscript presented in an intelligible fashion and written in standard English?

Reviewer #4: Yes

6. Review Comments to the Author

Reviewer #4: The paper is well written and the objectives are clear. The paper deals with the fabrication of dispensers made by a 3D printing process. The dispenser, developed during course of work, was compared with commercially available ones. Those developed here were less variable in volumes dispensed than were commercial ones used for comparison . This was attributed to the use of plasma and peg treatment which made the tips hydrophobic and the insides hydrophilic.

7. PLOS authors have the option to publish the peer review history of their article (what does this mean?). If published, this will include your full peer review and any attached files.

Reviewer #4: No

---

## [Editor Report · Acceptance letter]

11 Dec 2024

PONE-D-24-22915R1 

PLOS ONE

Dear Dr. Verma, 

I'm pleased to inform you that your manuscript has been deemed suitable for publication in PLOS ONE. Congratulations! Your manuscript is now being handed over to our production team.

Kind regards, 

on behalf of

Dr Enoch Aninagyei 

Academic Editor

PLOS ONE